# Physical Activity and Oxidative Stress in Aging

**DOI:** 10.3390/antiox13050557

**Published:** 2024-05-01

**Authors:** Rosamaria Militello, Simone Luti, Tania Gamberi, Alessio Pellegrino, Alessandra Modesti, Pietro Amedeo Modesti

**Affiliations:** 1Department of Experimental and Clinical Biomedical Sciences, University of Florence, 50134 Florence, Italy; rosamaria.militello@unifi.it (R.M.); simone.luti@unifi.it (S.L.); tania.gamberi@unifi.it (T.G.); 2Department of Experimental and Clinical Medicine, University of Florence, 50134 Florence, Italy; alessio.pellegrino@unifi.it (A.P.); pa.modesti@unifi.it (P.A.M.)

**Keywords:** aging, oxidative stress, sarcopenia, hormesis

## Abstract

Biological aging, characterized by changes in metabolism and physicochemical properties of cells, has an impact on public health. Environment and lifestyle, including factors like diet and physical activity, seem to play a key role in healthy aging. Several studies have shown that regular physical activity can enhance antioxidant defense mechanisms, including the activity of enzymes such as superoxide dismutase (SOD), catalase, and glutathione peroxidase. However, intense or prolonged exercise can also lead to an increase in reactive oxygen species (ROS) production temporarily, resulting in oxidative stress. This phenomenon is referred to as “exercise-induced oxidative stress”. The relationship between physical activity and oxidative stress in aging is complex and depends on various factors such as the type, intensity, duration, and frequency of exercise, as well as individual differences in antioxidant capacity and adaptation to exercise. In this review, we analyzed what is reported by several authors regarding the role of physical activity on oxidative stress in the aging process as well as the role of hormesis and physical exercise as tools for the prevention and treatment of sarcopenia, an aging-related disease. Finally, we reported what has recently been studied in relation to the effect of physical activity and sport on aging in women.

## 1. Introduction

The improvement in living conditions and advances in medicine led to an increase in life expectation. According to the World Health Organization (WHO) by 2050 people aged 60 years old and older will double from 12% to 22% reaching 2.1 billion while persons aged 80 years or older will triple, reaching 426 million [1].

Identifying strategies to maintain well-being in older age is important since the aging global population is a relatively new problem worldwide. Biological aging is characterized by changes in metabolism and physicochemical properties of cells that manifest with several complex health conditions having an impact on public health [2]. Environment and lifestyle, including factors like diet and physical activity, seem to play a key role in healthy aging.

The relationship between sporting activity and the generation of oxygen-free radicals with aging is complex. If the balance between physical activity and oxidative stress is disturbed by an excessive workload or by the presence of age-related metabolic alterations such as obesity or diabetes, cells, and organs may develop abnormalities due to an abnormal acceleration of the aging process [3]. Such abnormalities, also referred to as “exercise-induced oxidative stress”, may contribute to the development of chronic and degenerative disorders such as arthritis [4], autoimmune disorders [5], cardiovascular [6], and neurodegenerative diseases [7], inflammation, and cancer [8].

On the other hand, it is now clear that maintaining a correct physical activity program generates a moderate and short-term increase in free radicals, which can activate molecular mechanisms useful for the cell to adapt and improve the immunological defenses of the organism [8]. The extent to which reactive species are helpful or harmful depends on the exercise duration, intensity, fitness condition, and nutritional status of the individual [9]. Therefore, subjects at any age, with particular attention to aging, can benefit from constant, therefore repeated over time, physical activity to counteract the negative effects and toxicity of oxidative stress on health. Regular exercise improves antioxidant defenses and reduces lipid peroxidation levels both in adults and in aged individuals [10]. It alleviates the negative effects caused by free radicals [11]. Moreover, it offers many health benefits, including reduced risk of all-cause mortality, sarcopenia in the skeletal muscle, chronic disease, and premature death in elderly people [12]. Given that most of the reviews that have considered these aspects have largely oriented the discussion toward male-related issues, the purpose of this review is to direct attention to some specific problems of women.

Understanding the interplay between physical activity, oxidative stress, and aging has important clinical implications for the development of interventions aimed at promoting healthy aging and preventing age-related diseases. Strategies such as regular exercise, antioxidant supplementation, and lifestyle modifications can potentially help mitigate the negative effects of oxidative stress and improve overall health and longevity in older adults.

## 2. Aging

Aging is a complex process characterized by a gradual decline in physiological function and an increased vulnerability to diseases. It encompasses various physical, psychological, and social changes that occur as individuals grow older. Biological aging is influenced by genetic factors, environmental factors, lifestyle choices, and other variables. Some common manifestations of aging include wrinkles, reduced muscle mass, decreased bone density, diminished sensory functions, and cognitive decline. Oxidative stress, which results from an imbalance between the production of reactive oxygen species (ROS) and the body’s antioxidant defenses, is considered one of the key mechanisms underlying aging. Moreover, biological aging is a complex process characterized by the accumulation of DNA damage, cellular senescence, mitochondrial dysfunction, reduction in telomere length, loss of proteostasis, imbalanced metabolism, and stem cell exhaustion (Figure 1a) [13]. All these hallmarks lead to a progressive decline in tissue and organ function. Differences in aging between men and women encompass various biological and physiological effects shown in Figure 1b and in several points reported below [14].

Among biological factors, hormonal changes are due to menopause experienced by women which leads to decreased estrogen levels that can influence bone density [15], skin elasticity [16], and other physiological functions. On the contrary, men showed a testosterone decline, they experience a gradual decline in testosterone levels with age, which can affect muscle mass, bone density, and sexual function [17].

Among physiological differences, authors reported the following:*Muscle Mass*: Generally, men have greater muscle mass and strength compared to women due to higher testosterone levels, although this difference tends to diminish with age [18].*Fat Distribution*: Women typically have more subcutaneous fat, particularly around the hips and thighs, while men tend to accumulate more visceral fat around the abdomen, which can increase the risk of cardiovascular diseases [19].*Bone Density*: Women are at a higher risk of osteoporosis due to hormonal changes post-menopause, leading to decreased bone density and increased susceptibility to fractures [15].*Cognitive Decline*: While both men and women experience cognitive decline with age, research suggests that women may have a slightly lower risk of developing cognitive impairment and Alzheimer’s disease compared to men [20].

### 2.1. Oxidative Stress in Aging

Oxidative stress occurs when there is an imbalance between the production of ROS and the body’s ability to detoxify them or repair the resulting damage. ROS are highly reactive molecules that can damage cellular components such as proteins, lipids, and DNA, leading to cellular dysfunction and contributing to the aging process [21].

Specific actions of ROS depend on their chemical nature and the cellular context in which they are produced. Superoxide anion is produced primarily by the electron transport chain in mitochondria, and it directly damages cellular components [22] or can be converted into hydrogen peroxide that can diffuse across cell membranes. It can be a signaling molecule in various cellular processes or be detoxified by antioxidant enzymes such as catalase and glutathione peroxidase [23]. The most reactive molecule is hydroxyl radical formed through the Fenton reaction, involving the interaction of hydrogen peroxide with iron or copper and it is responsible for damaging DNA, proteins, and lipids [24] Although ROS play important roles in cell signaling and homeostasis, their accumulation can cause DNA strand breaks, base modifications, and cross-linking, leading to mutations and genomic instability, potentially leading to cell death or carcinogenesis [25]. ROS can also oxidize several amino acid residues in proteins, and this can disrupt enzymatic activity, protein–protein interactions, and cellular signaling pathways [26]. ROS can initiate lipid peroxidation, a chain reaction that can disrupt membrane integrity and lead to cell dysfunction [26]. Overall, while ROS plays essential roles in physiological processes, their excessive accumulation can lead to cellular damage and contribute to the pathogenesis of various diseases.

In aging, several factors contribute to increased oxidative stress. Several of these are summarized below and in Figure 2.

*Decline in Antioxidant Defenses*: The body’s antioxidant defenses, including enzymes like superoxide dismutase (SOD), catalase, and glutathione peroxidase, decline with age. This reduction in antioxidant capacity makes older individuals more susceptible to oxidative damage [27].*Mitochondrial Dysfunction*: Mitochondria, the energy-producing organelles within cells, are a major source of ROS. As mitochondria age, they become less efficient at producing energy and more prone to generating ROS as a by-product of respiration. This contributes to a vicious cycle of oxidative stress and mitochondrial dysfunction [28].*Inflammation*: Chronic low-grade inflammation, often termed “inflammaging”, is a hallmark of aging. Inflammatory processes can stimulate the production of ROS and exacerbate oxidative stress. Conversely, oxidative stress can also promote inflammation, creating a feedback loop that contributes to age-related tissue damage [29].*Accumulation of Damage*: Over time, cumulative oxidative damage to cellular components like DNA, proteins, and lipids can impair cellular function and contribute to age-related decline in tissue and organ function [30].The consequences of oxidative stress in aging are wide-ranging and can affect various physiological systems reported here:*Cellular Aging*: Oxidative damage contributes to cellular senescence, a state of irreversible growth arrest that limits the replicative capacity of cells and contributes to tissue aging [31].*Tissue Dysfunction*: Oxidative stress plays a role in age-related degenerative diseases such as cardiovascular disease, neurodegenerative diseases (e.g., Alzheimer’s disease, Parkinson’s disease), and age-related macular degeneration [32,33,34,35].*Cancer*: While oxidative stress can promote DNA damage and increase the risk of mutations that lead to cancer, paradoxically, cancer cells often exhibit increased antioxidant defenses to protect themselves from oxidative damage [36,37].*Immune Function*: Oxidative stress can impair immune function, making older individuals more susceptible to infections and less responsive to vaccines [38,39].

**Figure 2 antioxidants-13-00557-f002:**
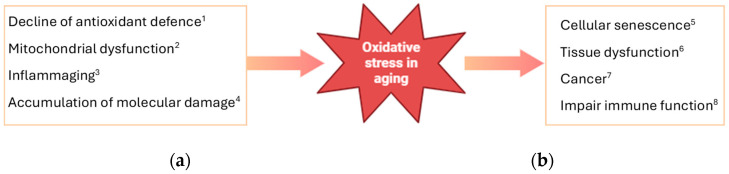
Oxidative stress in aging: (**a**) factors producing ROS in aging 1 [27]; 2 [28]; 3 [29]; 4 [30]; (**b**) consequences of oxidative stress in aging 5 [31]; 6 [32]; 7 [36]; 8 [38].

Efforts to mitigate oxidative stress and its effects on aging include lifestyle interventions such as maintaining a balanced diet rich in antioxidants, regular exercise, avoiding tobacco smoke and excessive alcohol consumption, and managing chronic conditions like diabetes and hypertension. More than 300 theories have been proposed to explain the phenomena of aging but a single theory could not explain all the mechanisms of aging [30]. ROS are important for the defense mechanism and signaling transduction [40,41]. These pathways play an important role in various cellular processes such as cell growth, inflammatory response, autophagy, or adaptive response to oxidative stress [42]. However, accumulations of ROS have a negative effect on health and induce oxidative stress (OS) [43]. The primary endogenous sources for ROS production are mitochondria via oxidative phosphorylation but in mammals’ cells, different protein complexes are responsible for ROS production [44].

### 2.2. Hormesis, Aging and Physical Activity

Southam and Erhlich indicate the effect of plant extracts on fungi culture [45] used the term hormesis for the first time in the scientific literature in 1943. Hormesis is a biological phenomenon where exposure to low doses of a stressor or toxin induces adaptive responses in an organism, resulting in improved health and longevity. This concept suggests that mild stressors can activate cellular mechanisms that enhance resilience and resistance to more severe stressors. Exercise, for example, is one of the hormetic stressors that has been extensively studied for its beneficial effects on aging. In the context of aging, hormesis has been proposed as a potential mechanism to delay or mitigate age-related decline and promote longevity. Several stressors, such as calorie restriction, exercise, heat shock, and certain phytochemicals, have been studied for their hormetic effects on aging [46]. The hormetic effects of physical activity on aging are thought to involve multiple mechanisms, including the activation of cellular stress response pathways (such as the AMPK and sirtuin pathways) [47], enhanced autophagy [48], and improved mitochondrial function [49]. These adaptations help the body cope with stress more effectively, leading to greater resilience and longevity. Regular physical activity, particularly aerobic and resistance training, induces mild oxidative stress and inflammation, which triggers adaptive responses in muscles, bones, and other tissues, leading to improved function and resilience with age [50]. This stress triggers a cascade of physiological responses aimed at adapting to the demands placed upon it. These adaptations, shown in Figure 3, include:*Muscle Growth and Strength*: Resistance training, such as weightlifting, places stress on muscles, leading to microscopic damage to muscle fibers. In response, the body repairs and rebuilds these fibers, resulting in increased muscle mass and strength [51].*Cardiovascular Fitness*: Aerobic exercise, like jogging or cycling, stresses the cardiovascular system by increasing heart rate and blood flow. Over time, the heart becomes more efficient at pumping blood, and the blood vessels become more elastic, leading to improved cardiovascular fitness and reduced risk of heart disease [52].*Bone Density*: Weight-bearing exercises, such as walking or running, stress the bones, stimulating bone remodeling and increasing bone density. This helps prevent osteoporosis and reduces the risk of fractures [53,54].*Metabolic Health*: Exercise enhances insulin sensitivity, promotes glucose uptake by muscles, and improves lipid profiles, all of which contribute to better metabolic health and reduced risk of conditions like type 2 diabetes and metabolic syndrome [55,56].*Mood and Cognitive Function*: Physical activity triggers the release of endorphins and other neurotransmitters that promote feelings of well-being and reduce stress and anxiety. Regular exercise has also been linked to improved cognitive function and reduced risk of cognitive decline with aging [56,57]. It is important to note that while moderate exercise provides a beneficial hormetic effect, excessive or overly intense exercise can have the opposite effect, leading to excessive stress, inflammation, and potential injury.

**Figure 3 antioxidants-13-00557-f003:**
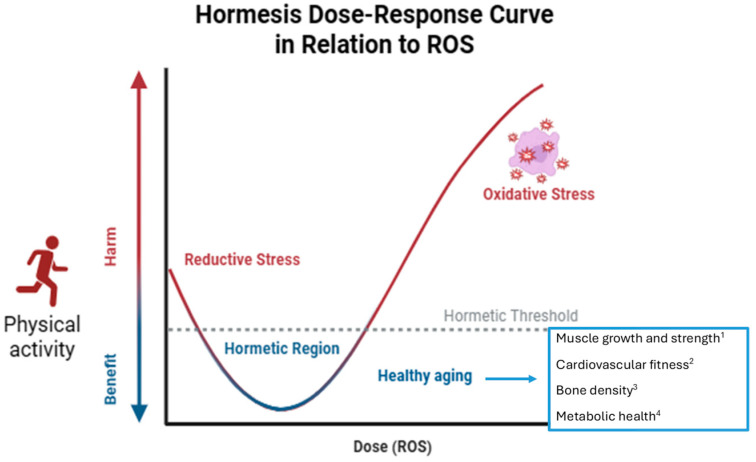
Exercise as a hormetic stressor and its beneficial effects: on muscle strength 1 [51]; cardiovascular fitness 2 [52]; bone density 3 [53] and metabolic health 4 [55] during aging.

Therefore, finding the right balance and incorporating rest and recovery into exercise routines is crucial for reaping the full benefits of hormesis from physical activity [49,58]. Overall, hormesis provides a theoretical framework for understanding how exposure to mild stressors can stimulate adaptive responses that promote health and longevity. While more research is needed to fully elucidate the mechanisms underlying hormesis and its implications for aging, the concept holds promise for developing interventions to delay age-related decline and extend lifespan.

### 2.3. Physical Activity in Aging

Regular physical activity has been shown to have numerous benefits for overall health and well-being, including mitigating some of the effects of aging. Exercise helps improve cardiovascular health, maintain muscle mass and strength, enhance cognitive function, and increase overall longevity [59]. Regular physical activity helps to prevent cardiovascular and metabolic disease, obesity, falls, cognitive impairments, osteoporosis, and muscular weakness are decreased by regularly completing activities ranging from low intensity walking through to more vigorous sports and resistance exercise. But despite evidence showing that it is safe for healthy and older people, the participation of the latter remains low.

In general, the more often a person is physically active, the better their physical capability. This is due to adaptations of physiological systems, most notably within the neuromuscular system to coordinate movements, the cardiopulmonary system to more effectively distribute oxygen and nutrients around the body, and metabolic processes particularly those regulating glucose and fatty acid metabolism, which collectively increase overall aerobic power and physical capability. Thus, the trajectory towards frailty is directly modifiable through physical activity habits [60,61]. Physical activity is extremely important for elderly individuals as it offers a wide range of physical and mental health benefits. Regular physical activity can help improve strength, flexibility, balance, and cardiovascular health, all of which are crucial for maintaining independence and reducing the risk of falls and injuries in older adults [62].

Engaging in physical activity can also help manage chronic conditions such as arthritis, osteoporosis, and diabetes. It can improve mood, reduce symptoms of depression and anxiety, and enhance cognitive function. Additionally, staying physically active promotes social interaction and can contribute to a greater sense of well-being and overall quality of life [63]. It is important for elderly individuals to choose activities that are appropriate for their fitness level and health status. This might include walking, swimming, cycling, tai chi, yoga, or strength training exercises [64]. It is also important to consult with a healthcare professional before starting any new exercise program, especially if there are underlying health concerns [65]. Overall, encouraging and supporting elderly individuals to remain physically active can significantly improve their health and vitality as they age [66].

## 3. Sarcopenia in Aging

Sarcopenia refers to the gradual loss of muscle mass, strength, and function that occurs with aging. It is a natural part of the aging process, typically starting around middle age but accelerating after the age of 75. Several factors contribute to sarcopenia, including decreased physical activity as reported in Figure 4 [67,68]. It is a multifactorial and complex phenomenon whose underlying mechanisms are not clearly defined. Based on etiological factors sarcopenia is classified as primary when no specific cause other than aging is evident and secondary when other factors like inflammatory and endocrine diseases are evident. Moreover, bad habits like physical inactivity and undernutrition may be a factor that contributes to the development of sarcopenia [69].

The European Working Group on Sarcopenia in Older People (EWGSOP) proposed diagnostic criteria based on muscle mass, muscle strength, and physical performance [70]. In 2018, EWGSOP revised the parameter of sarcopenia diagnosis giving more importance to muscle strength than muscle mass to predict adverse outcomes [71]. Lack of exercise is one of the major risk factors of sarcopenia and resistance exercise is the main non-pharmacological tool for the management of sarcopenia [72]. In aging muscle, there is an imbalance between protein synthesis and degradation that leads to a decrease in muscle mass [73]. The consequences of sarcopenia can be significant, as it can lead to decreased mobility, increased risk of falls and fractures, loss of independence, and overall decreased quality of life. It has also been associated with increased mortality rates [74].

Preventing and managing sarcopenia involves a multifaceted approach that includes regular physical activity, especially resistance training to maintain muscle mass and strength, adequate protein intake to support muscle repair and growth, and overall good nutrition [75]. Additionally, managing chronic illnesses and addressing hormonal imbalances can help mitigate the effects of sarcopenia [76,77].

Early detection and intervention are crucial in combating sarcopenia. Regular assessments of muscle mass, strength, and function can help identify individuals at risk, allowing timely interventions to preserve muscle health and function as people age [78].

**Figure 4 antioxidants-13-00557-f004:**
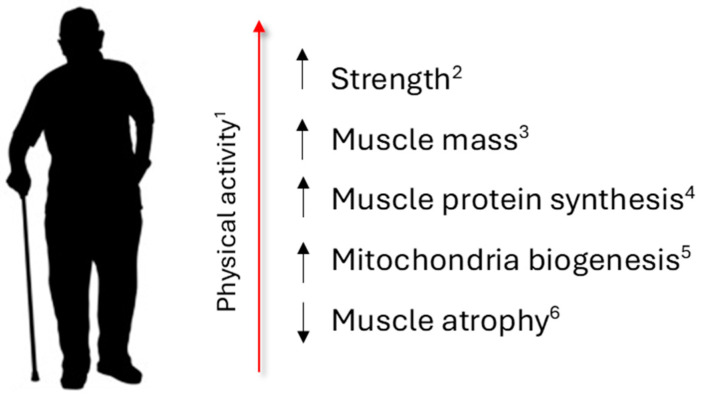
Regular physical activity 1 [63] prevents and manages sarcopenia increasing (arrows up) strength 2 [70]; muscle mass 3 [71]; muscle protein synthesis 4 [72]; mitochondrial biogenesis 5 [78]; and reducing (arrow down) muscle atrophy 5 [79].

### 3.1. Sarcopenia, Mitochondrial Dysfunction and Physical Activity

Different studies have highlighted the link between mitochondrial dysfunction and sarcopenia. In aging muscle, evident mitochondrial alterations like morphological changes and downregulation of PGC1-alpha, are the factors that regulate mitochondrial biogenesis [79]. Otherwise, mitochondrial damage induces the accumulation of ROS [80]. Furthermore, PGC-1α expression stimulates the expression of antioxidant genes, including heme oxygenase 1 (HO-1) [81]. The imbalance between the production of ROS, Reactive Nitrogen Species (RNS), and antioxidant defenses in the body is an early biomarker of sarcopenia [82]. ROS are produced during muscle contraction by the mitochondrial electron transport chain during normal oxidative respiration [83]. With aging, skeletal muscle increases the production of ROS, which contributes to increased damage of cells and muscle atrophy [84] because enhancing the ubiquitin–proteasome system, resulting in skeletal muscle atrophy [85]. Progressive resistance training is the most studied method of exercise. Aerobic exercise otherwise contributes to activating mitochondria biogenesis [86] and increases the synthesis of muscle proteins like myostatin whose mRNA expression increases [87]. In general, aerobic exercise tends to ameliorate mitochondrial-associated problems. In older adults, no particular type of exercise satisfies all the needs requested for the use of exercise as a therapeutic tool in age-related sarcopenia, and thus well-rounded aerobic and resistance exercise programs are recommended [88]. Moreover, vibration therapy in case of inability is able to improve physical measurement [89].

### 3.2. Aging and Physical Activity in Women

Aging in females involves a complex interplay of physiological, psychological, and social changes that occur, as women grow older. Physical activity plays a crucial role in mitigating many of the physiological and psychological changes associated with aging in females. Overall, aging in females is a multifaceted process influenced by various factors. Here and in Figure 5, some key aspects of aging in females and how physical activity can positively influence them are summarized.

*Menopause*: Characterized by a decline in hormone levels, particularly estrogen and progesterone, which can lead to various symptoms such as hot flashes, mood swings, weight gain, decreased bone density, and increased risk of cardiovascular disease. Regular physical activity can help alleviate many of these symptoms and improve overall well-being during menopause [90].*Bone Health*: Women are at a higher risk of osteoporosis compared to men, particularly after menopause. Osteoporosis is characterized by decreased bone density and an increased risk of fractures. Maintaining regular exercise is crucial for preserving bone health, weight-bearing and resistance exercises help to maintain bone density and strength, reducing the risk of osteoporosis and fractures. Activities such as walking, jogging, dancing, and weightlifting are beneficial for bone health [91,92]. Moreover, several authors reported that resistance training exercises help to build and maintain muscle mass, strength, and function, which can decline with age. Strong muscles support joint health, balance, and mobility, reducing the risk of falls and injuries [93].*Weight Management and Metabolic Changes*: Menopause often brings about weight gain, especially around the abdomen. The metabolic rate tends to decrease with age, making weight management more challenging. Regular physical activity helps to maintain a healthy weight, improve insulin sensitivity, and reduce the risk of metabolic conditions such as type 2 diabetes and cardiovascular disease. Both aerobic exercise (e.g., walking, swimming, cycling) and strength training contribute to metabolic health [94,95].*Cardiovascular Health*: Post-menopausal women have an increased risk of cardiovascular disease, partly due to changes in hormone levels [96,97]. Managing risk factors such as high blood pressure, high cholesterol, and diabetes through lifestyle modifications and medical treatment is essential for maintaining heart health [98]. Aerobic exercises such as brisk walking, cycling, and swimming are effective for improving cardiovascular fitness, reducing blood pressure, lowering cholesterol levels, and decreasing the risk of heart disease and stroke [99,100].*Cognitive Health, Psychological, and Social Well-being*: While cognitive decline can occur with age, women may have a lower risk of developing certain neurodegenerative diseases such as Alzheimer’s disease compared to men [101]. However, they may experience cognitive changes associated with hormonal fluctuations and aging [102]. Moreover, aging can bring about psychological and social changes, including shifts in roles and relationships, retirement, caregiving responsibilities, and coping with loss [103]. Physical activity improves mood, reduces symptoms of anxiety and depression, and enhances cognitive function and brain health in older adults. Engaging in regular exercise can boost self-esteem, promote relaxation, and alleviate stress [104,105].Moreover, participating in group-based physical activities or exercise classes can provide opportunities for social interaction, connection, and support, which are important for overall well-being and quality of life, particularly as women age [106,107,108]. It is important for women to choose activities that they enjoy and that fit their individual preferences, abilities, and health goals. Consulting with a healthcare provider or a fitness professional can help in developing a safe and personalized exercise program tailored to individual needs and medical conditions.*Functional Independence*: Maintaining physical fitness and functional abilities through regular exercise supports independence in activities of daily living, such as dressing, bathing, cooking, and household chores, allowing women to maintain autonomy and quality of life as they age [109,110,111,112].*Joint Health and Flexibility*: Activities that promote joint mobility and flexibility, such as yoga, Pilates, and stretching exercises, help to maintain joint health and range of motion, reducing the risk of stiffness and arthritis-related pain [113,114,115,116].

**Figure 5 antioxidants-13-00557-f005:**
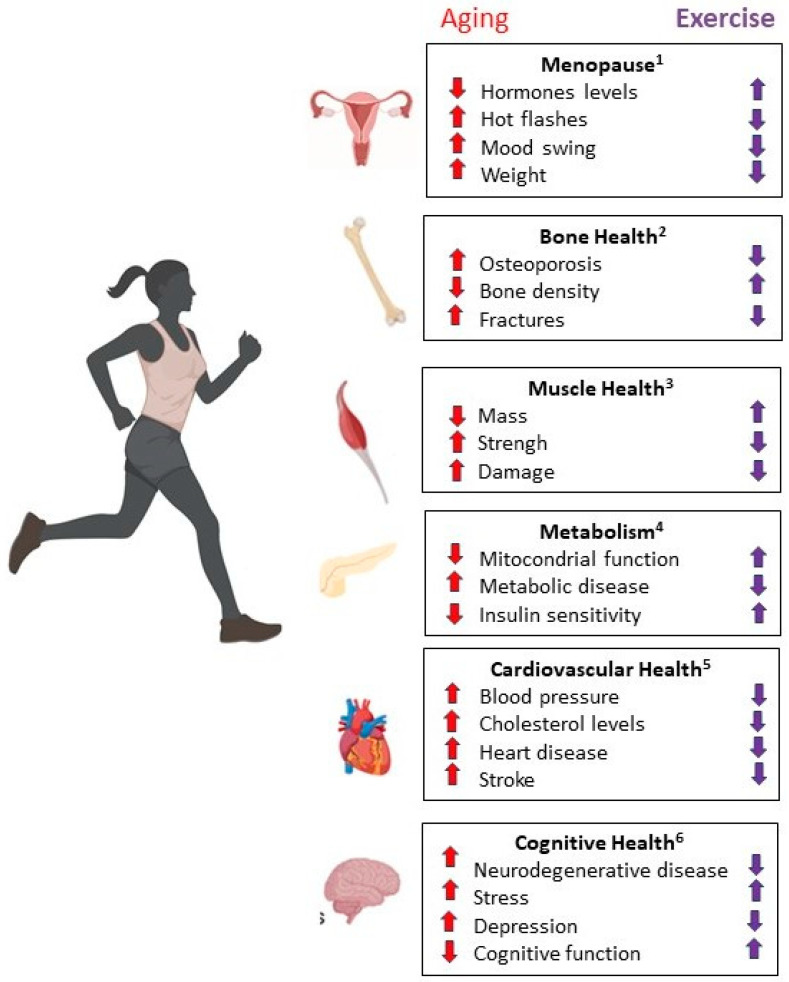
Aging in females and positive influence of physical activity. Changes that occur in the female organism during aging (red arrows up—increase; red arrows down decrease); positive changes due to regular physical activity. 1 [90]; 2 [91,92]; 3 [93]; 4 [94,95]; 5 [96,97,98]; 6 [101,102].

### 3.3. Sport and Elderly Women

Regular participation in sports helps women maintain physical fitness, strength, flexibility, and cardiovascular health as they age (Figure 6). Activities such as running, swimming, cycling, tennis, and team sports provide opportunities for aerobic exercise, muscle strengthening, and agility training, contributing to overall health and vitality.

Moreover, sports participation can enhance self-esteem, self-confidence, and a sense of empowerment in women of all ages [117]. Achieving personal fitness goals, mastering new skills, and overcoming challenges in sports can build resilience and a positive self-image, particularly as societal attitudes toward aging continue to evolve. Engaging in sports offers opportunities for continuous learning, skill development, and personal growth throughout the aging process [118]. Women can explore new sports, adapt their training routines, and set new goals to stay active and motivated as they navigate different life stages [119]. It is important for women to choose sports and physical activities that align with their interests, abilities, and health goals, and to consult with healthcare professionals or fitness experts as needed to ensure safety and appropriate exercise programming. With the right support and resources, sports participation can enrich the lives of women as they age, promoting health, vitality, and overall well-being.

## 4. Conclusions

What is reported in this review suggests that normal daily activity in elderly subjects preserves the independence of the elderly and keeps their oxidative state at low levels. Moreover, encouraging physical activity and sports practice among the elderly, regardless of gender, should be a priority in healthcare and community settings. Tailored exercise programs, accessibility to suitable facilities, social support networks, and education on the importance of staying active can all contribute to improving the physical and mental well-being of elderly men and women. Additionally, addressing specific barriers faced by each gender group can help in promoting inclusivity and participation in physical activities among the elderly population.

## Figures and Tables

**Figure 1 antioxidants-13-00557-f001:**
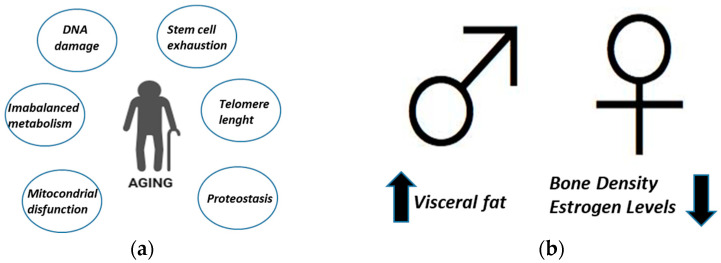
Biological aging. (**a**) Different factors influencing aging [4]; (**b**) differences in aging between men and women. In men, there is an increase in visceral fat (arrow up) while in women there is a decrease in bone density and estrogen levels (arrow down).

**Figure 6 antioxidants-13-00557-f006:**
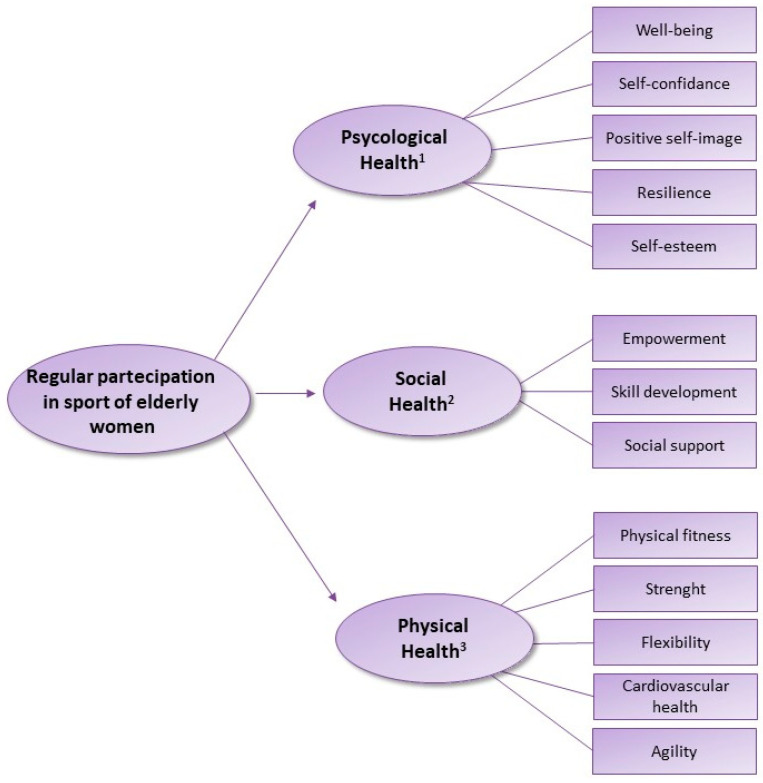
Effects of participation in sports on physical fitness, strength, flexibility, and cardiovascular health in aged women, 1 [104,105]; 2 [117]; 3 [118,119].

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
