# Peer review of "Physical Activity and Oxidative Stress in Aging"

_antioxidants, 2024, doi:10.3390/antiox13050557_

Round 1
Reviewer 1 Report
The current report has an emphasis on the effects of an altered anti-oxidant system involving enzymes such as SOD, catalase and glutathione peroxidase. Such abnormalities are also referred to as “exercise-induced oxidative stress.” The authors emphasize the importance linking such responses to physicality. It would be helpful if the authors briefly linked these data together and also pointed out how aging can disrupt some of the potential harmful outcomes.
This is a review of the literature as it relates to physical activity in human adults and oxidative stress in aging. It is known that if the balance between physical activity and oxidative stress of aging is disturbed, cells and organs may develop abnormalities. The current report has an emphasis on the effects of an altered anti-oxidant system involving enzymes such as SOD, catalase and glutathione peroxidase. Such abnormalities are also referred to as “exercise-induced oxidative stress.” The authors emphasize the importance linking such responses to physicality. In general, all of this information is slowly coming together. It would be helpful if the authors briefly linked these data together and also pointed out how aging can disrupt some of the potential harmful outcomes.
Author Response
Reviewer 1
Major comments
The current report has an emphasis on the effects of an altered anti-oxidant system involving enzymes such as SOD, catalase and glutathione peroxidase. Such abnormalities are also referred to as “exercise-induced oxidative stress.” The authors emphasize the importance linking such responses to physicality. It would be helpful if the authors briefly linked these data together and also pointed out how aging can disrupt some of the potential harmful outcomes.
Detail comments
This is a review of the literature as it relates to physical activity in human adults and oxidative stress in aging. It is known that if the balance between physical activity and oxidative stress of aging is disturbed, cells and organs may develop abnormalities. The current report has an emphasis on the effects of an altered anti-oxidant system involving enzymes such as SOD, catalase and glutathione peroxidase. Such abnormalities are also referred to as “exercise-induced oxidative stress.” The authors emphasize the importance linking such responses to physicality. In general, all of this information is slowly coming together. It would be helpful if the authors briefly linked these data together and also pointed out how aging can disrupt some of the potential harmful outcomes.
Thank you for highlighting a very important aspect that perhaps was not clearly expressed in the first draft. The following text replaces lines 35-47 in the Introduction section:
“The relationship between sporting activity and the generation of oxygen free radicals with aging is complex. If the balance between physical activity and oxidative stress is disturbed by an excessive workload or by the presence of age-related metabolic alterations such as obesity or diabetes, cells and organs may develop abnormalities due to an abnormal acceleration of the aging process (Carru C, Da Boit M, Paliogiannis P, Zinellu A, Sotgia S, Sibson R, Meakin JR, Aspden RM, Mangoni AA, Gray SR. Markers of oxidative stress, skeletal muscle mass and function, and their responses to resistance exercise training in older adults. Exp Gerontol. 2018 Mar;103:101-106. doi: 10.1016/j.exger.2017.12.024. Epub 2018 Jan 9. PMID: 29326089.). Such abnormalities, also referred to as “exercise-induced oxidative stress”, may contribute in the development of chronic and degenerative disorders such as arthritis (Phull AR, Nasir B, Haq IU, Kim SJ. Oxidative stress, consequences and ROS mediated cellular signaling in rheumatoid arthritis. Chem Biol Interact. 2018 Feb 1;281:121-136. doi: 10.1016/j.cbi.2017.12.024. Epub 2017 Dec 16. PMID: 29258867.), autoimmune disorders (Kaffe ET, Rigopoulou EI, Koukoulis GK, Dalekos GN, Moulas AN. Oxidative stress and antioxidant status in patients with autoimmune liver diseases. Redox Rep. 2015 Jan;20(1):33-41. doi: 10.1179/1351000214Y.0000000101. Epub 2014 Aug 13. PMID: 25117650; PMCID: PMC6837668.), cardiovascular (Senoner T, Dichtl W. Oxidative Stress in Cardiovascular Diseases: Still a Therapeutic Target? Nutrients. 2019 Sep 4;11(9):2090. doi: 10.3390/nu11092090. PMID: 31487802; PMCID: PMC6769522.) and neurodegenerative diseases (Kim GH, Kim JE, Rhie SJ, Yoon S. The Role of Oxidative Stress in Neurodegenerative Diseases. Exp Neurobiol. 2015 Dec;24(4):325-40. doi: 10.5607/en.2015.24.4.325. Epub 2015 Oct 12. PMID: 26713080; PMCID: PMC4688332.), inflammation and cancer (Simioni C, Zauli G, Martelli AM, Vitale M, Sacchetti G, Gonelli A, Neri LM. Oxidative stress: role of physical exercise and antioxidant nutraceuticals in adulthood and aging. Oncotarget. 2018 Mar 30;9(24):17181-17198. doi: 10.18632/oncotarget.24729. PMID: 29682215; PMCID: PMC5908316.).
On the other hand, it is now clear that maintaining a correct physical activity program generates a moderate and short-term increase in free radicals, which can activate molecular mechanisms useful for the cell to adapt and improve the immunological defenses of the organism (Simioni C, Zauli G, Martelli AM, Vitale M, Sacchetti G, Gonelli A, Neri LM. Oxidative stress: role of physical exercise and antioxidant nutraceuticals in adulthood and aging. Oncotarget. 2018 Mar 30;9(24):17181-17198. doi: 10.18632/oncotarget.24729. PMID: 29682215; PMCID: PMC5908316). The extent to which reactive species are or helpful or harmful depends on the exercise duration, intensity, fitness condition and nutritional status of the individual (Margonis K, Fatouros IG, Jamurtas AZ, Nikolaidis MG, Douroudos I, Chatzinikolaou A, Mitrakou A, Mastorakos G, Papassotiriou I, Taxildaris K, Kouretas D. Oxidative stress biomarkers responses to physical overtraining: implications for diagnosis. Free Radic Biol Med. 2007 Sep 15;43(6):901-10. doi: 10.1016/j.freeradbiomed.2007.05.022. Epub 2007 May 23. Erratum in: Free Radic Biol Med. 2008 Jun 15;44(12):2058. PMID: 17697935.). Therefore, subjects at any age, with particular attention to aging, can benefit from constant, therefore repeated over time, physical activity to counteract the negative effects and toxicity of oxidative stress on health. Regular exercise improves antioxidant defenses and lowers lipid peroxidation levels both in adult and in aged individuals (Bouzid MA, Filaire E, Matran R, Robin S, Fabre C. Lifelong Voluntary Exercise Modulates Age-Related Changes in Oxidative Stress. Int J Sports Med. 2018 Jan;39(1):21-28. doi: 10.1055/s-0043-119882. Epub 2017 Nov 23. PMID: 29169189.), alleviates the negative effects caused by free radicals (Condello G, Ling FC, Bianco A, Chastin S, Cardon G, Ciarapica D, Conte D, Cortis C, De Craemer M, Di Blasio A, Gjaka M, Hansen S, Holdsworth M, Iacoviello L, Izzicupo P, Jaeschke L, Leone L, Manoni L, Menescardi C, Migliaccio S, Nazare JA, Perchoux C, Pesce C, Pierik F, Pischon T, Polito A, Puggina A, Sannella A, Schlicht W, Schulz H, Simon C, Steinbrecher A, MacDonncha C, Capranica L; DEDIPAC consortium. Using concept mapping in the development of the EU-PAD framework (EUropean-Physical Activity Determinants across the life course): a DEDIPAC-study. BMC Public Health. 2016 Nov 9;16(1):1145. doi: 10.1186/s12889-016-3800-8. PMID: 27825370; PMCID: PMC5101801.) and offers many health benefits, including reduced risk of all-cause mortality, sarcopenia in the skeletal muscle, chronic disease, and premature death in elderly people (Golbidi S, Badran M, Laher I. Antioxidant and anti-inflammatory effects of exercise in diabetic patients. Exp Diabetes Res. 2012;2012:941868. doi: 10.1155/2012/941868. Epub 2011 Oct 11. PMID: 22007193; PMCID: PMC3191828.).
Previous reviews that have considered these aspects have largely oriented the discussion towards male-related issues. The purpose of this review is to direct attention to the more specific problems of women.”

Reviewer 2 Report
The authors summarized the role of physical activity on oxidative stress in aging process as well as the role of hormesis and physical exercise as tools for prevention and treatment of sarcopenia. The impacts of physical activity on aging in women are also being highlighted. The review article summarizes the current knowledge from a specific and interesting perspective, and may arouse interest to those with specific interest in this area.
The authors summarized the role of physical activity on oxidative stress in aging process as well as the role of hormesis and physical exercise as tools for prevention and treatment of sarcopenia. The impacts of physical activity on aging in women are also being highlighted.
1. The authors are suggested to supplement more references when making statement. For example, from line 72-88, no reference is included.
2. ROS refers to a group of molecules, such as H2O2, superoxide, etc… A short summary on the common or specific actions of ROS shall be mentioned.
3. It is not justified why aging in women is discussed in more detail, but not men?
4. The word “ageing” and “aging” shall be unified throughout the manuscript.
5. In Figure 1, “lenght” is a typo.
6. The manuscript need to be proof-read again. Several typos are observed.
Author Response
Reviewer 2
Major comments
The authors summarized the role of physical activity on oxidative stress in aging process as well as the role of hormesis and physical exercise as tools for prevention and treatment of sarcopenia. The impacts of physical activity on aging in women are also being highlighted. The review article summarizes the current knowledge from a specific and interesting perspective, and may arouse interest to those with specific interest in this area.
The authors summarized the role of physical activity on oxidative stress in aging process as well as the role of hormesis and physical exercise as tools for prevention and treatment of sarcopenia. The impacts of physical activity on aging in women are also being highlighted.
- The authors are suggested to supplement more references when making statement. For example, from line 72-88, no reference is included.
You are right, new references were added. The text was modified as follows (lines 70-88):
“Among biological factors, hormonal changes are due to menopause experienced by women that leads to decreased estrogen levels that can influence bone density (Ji MX, Yu Q. Primary osteoporosis in postmenopausal women. Chronic Dis Transl Med. 2015 Mar 21;1(1):9-13. doi: 10.1016/j.cdtm.2015.02.006. PMID: 29062981; PMCID: PMC5643776.), skin elasticity (Calleja-Agius J, Brincat M. The effect of menopause on the skin and other connective tissues. Gynecol Endocrinol. 2012 Apr;28(4):273-7. doi: 10.3109/09513590.2011.613970. Epub 2011 Oct 4. PMID: 21970508.), and other physiological functions. On the contrary, men showed a testosterone decline, they experience a gradual decline in testosterone levels with age, which can affect muscle mass, bone density, and sexual function (Zirkin BR, Tenover JL. Aging and declining testosterone: past, present, and hopes for the future. J Androl. 2012 Nov-Dec;33(6):1111-8. doi: 10.2164/jandrol.112.017160. Epub 2012 Aug 9. PMID: 22879528; PMCID: PMC4077344.).
Among physiological differences authors reported:
Muscle Mass: Generally, men have greater muscle mass and strength compared to women due to higher testosterone levels, although this difference tends to diminish with age (Handelsman DJ, Hirschberg AL, Bermon S. Circulating Testosterone as the Hormonal Basis of Sex Differences in Athletic Performance. Endocr Rev. 2018 Oct 1;39(5):803-829. doi: 10.1210/er.2018-00020. PMID: 30010735; PMCID: PMC6391653).
Fat Distribution: Women typically have more subcutaneous fat, particularly around the hips and thighs, while men tend to accumulate more visceral fat around the abdomen, which can increase the risk of cardiovascular diseases (Palmer BF, Clegg DJ. The sexual dimorphism of obesity. Mol Cell Endocrinol. 2015 Feb 15;402:113-9. doi: 10.1016/j.mce.2014.11.029. Epub 2015 Jan 8. PMID: 25578600; PMCID: PMC4326001).
Bone Density: Women are at a higher risk of osteoporosis due to hormonal changes post-menopause, leading to decreased bone density and increased susceptibility to fractures (Ji MX, Yu Q. Primary osteoporosis in postmenopausal women. Chronic Dis Transl Med. 2015 Mar 21;1(1):9-13. doi: 10.1016/j.cdtm.2015.02.006. PMID: 29062981; PMCID: PMC5643776.).
Cognitive Decline: While both men and women experience cognitive decline with age, research suggests that women may have a slightly lower risk of developing cognitive impairment and Alzheimer's disease compared to men (Podcasy JL, Epperson CN. Considering sex and gender in Alzheimer disease and other dementias. Dialogues Clin Neurosci. 2016 Dec;18(4):437-446. doi: 10.31887/DCNS.2016.18.4/cepperson. PMID: 28179815; PMCID: PMC5286729.).”
- ROS refers to a group of molecules, such as H2O2, superoxide, etc… A short summary on the common or specific actions of ROS shall be mentioned.
You are right, new references were added and the chapter 2.2 Oxidative stress in aging, was modified in as follow:
“Specific actions of ROS depend on their chemical nature and the cellular context in which they are produced. Superoxide Anion is produced primarily by the electron transport chain in mitochondria, and it directly damage cellular components (Cáceres L, Paz ML, Garcés M, Calabró V, Magnani ND, Martinefski M, Martino Adami PV, Caltana L, Tasat D, Morelli L, Tripodi V, Valacchi G, Alvarez S, González Maglio D, Marchini T, Evelson P. NADPH oxidase and mitochondria are relevant sources of superoxide anion in the oxinflammatory response of macrophages exposed to airborne particulate matter. Ecotoxicol Environ Saf. 2020 Dec 1;205:111186. doi: 10.1016/j.ecoenv.2020.111186. Epub 2020 Aug 24. PMID: 32853868.) or can converted in hydrogen peroxide that can diffuse across cell membranes. It can be a signaling molecule in various cellular processes or be detoxified by antioxidant enzymes such as catalase and glutathione peroxidase (Sies H. Hydrogen peroxide as a central redox signaling molecule in physiological oxidative stress: Oxidative eustress. Redox Biol. 2017 Apr;11:613-619. doi: 10.1016/j.redox.2016.12.035. Epub 2017 Jan 5. PMID: 28110218; PMCID: PMC5256672.). The the most reactive molecule is hydroxyl radical formed through the Fenton reaction, involving the interaction of hydrogen peroxide with iron or copper and il is responsible for damaging DNA, proteins, and lipids (Jakubczyk K, Dec K, KaÅ‚duÅ„ska J, Kawczuga D, Kochman J, Janda K. Reactive oxygen species - sources, functions, oxidative damage. Pol Merkur Lekarski. 2020 Apr 22;48(284):124-127. PMID: 32352946.). Although ROS play important roles in cell signaling and homeostasis, their accumulation can cause DNA strand breaks, base modifications, and cross-linking, leading to mutations and genomic instability, potentially leading to cell death or carcinogenesis (Khan F, Ali A, Ali R. Enhanced recognition of hydroxyl radical modified plasmid DNA by circulating cancer antibodies. J Exp Clin Cancer Res. 2005 Jun;24(2):289-96. PMID: 16110763.). ROS can also oxidize several amino acid residues in proteins and this can disrupt enzymatic activity, protein-protein interactions, and cellular signaling pathways (Zeeshan HM, Lee GH, Kim HR, Chae HJ. Endoplasmic Reticulum Stress and Associated ROS. Int J Mol Sci. 2016 Mar 2;17(3):327. doi: 10.3390/ijms17030327. PMID: 26950115; PMCID: PMC4813189.). ROS can initiate lipid peroxidation, a chain reaction which can disrupt membrane integrity and lead to cell dysfunction (Zeeshan HM, Lee GH, Kim HR, Chae HJ. Endoplasmic Reticulum Stress and Associated ROS. Int J Mol Sci. 2016 Mar 2;17(3):327. doi: 10.3390/ijms17030327. PMID: 26950115; PMCID: PMC4813189.). Overall, while ROS play essential roles in physiological processes, their excessive accumulation can lead to cellular damage and contribute to the pathogenesis of various diseases.”
- It is not justified why aging in women is discussed in more detail, but not men?
Previous reviews that have considered issues related to physical activity and oxidative stress in aging have largely oriented the discussion towards male-related issues. The purpose of this review is to direct attention to the more specific problems of women.
4 The word “ageing” and “aging” shall be unified throughout the manuscript.
You agree and we unified.
- In Figure 1, “length” is a typo.
We don't understand what the reviewer means, the word length is correct.
The manuscript need to be proof-read again. Several typos are observed.
Thank you for the comment, we have reread the manuscript carefully and hope to have corrected all the typos

Round 2
Reviewer 2 Report
All the issues have been addressed.
All the issues have been addressed.